# Coinfection with Yellow Head Virus Genotype 8 (YHV-8) and Oriental Wenrivirus 1 (OWV1) in Wild *Penaeus chinensis* from the Yellow Sea

**DOI:** 10.3390/v15020361

**Published:** 2023-01-27

**Authors:** Jiahao Qin, Fanzeng Meng, Guohao Wang, Yujin Chen, Fan Zhang, Chen Li, Xuan Dong, Jie Huang

**Affiliations:** 1College of Fisheries and Life Science, Shanghai Ocean University, Shanghai 201306, China; 2Yellow Sea Fisheries Research Institute, Chinese Academy of Fishery Sciences, Laboratory for Marine Fisheries Science and Food Production Processes, Pilot National Laboratory for Marine Science and Technology (Qingdao), Key Laboratory of Maricultural Organism Disease Control, Ministry of Agriculture and Rural Affairs, Qingdao Key Laboratory of Mariculture Epidemiology and Biosecurity, Qingdao 266071, China; 3College of Fisheries and Life Science, Dalian Ocean University, Dalian 116023, China; 4College of Marine Science and Biological Engineering, Qingdao University of Science and Technology, Qingdao 266110, China; 5Network of Aquaculture Centres in Asia-Pacific, Bangkok 10900, Thailand

**Keywords:** YHV-8, OWV1, IHHNV, DIV1, *Penaeus chinensis*, coinfection

## Abstract

At present, there are few studies on the epidemiology of diseases in wild Chinese white shrimp *Penaeus chinensis*. In order to enrich the epidemiological information of the World Organisation for Animal Health (WOAH)-listed and emerging diseases in wild *P. chinensis*, we collected a total of 37 wild *P. chinensis* from the Yellow Sea in the past three years and carried out molecular detection tests for eleven shrimp pathogens. The results showed that infectious hypodermal and hematopoietic necrosis virus (IHHNV), Decapod iridescent virus 1 (DIV1), yellow head virus genotype 8 (YHV-8), and oriental wenrivirus 1 (OWV1) could be detected in collected wild *P. chinensis*. Among them, the coexistence of IHHNV and DIV1 was confirmed using qPCR, PCR, and sequence analysis with pooled samples. The infection with YHV-8 and OWV1 in shrimp was studied using molecular diagnosis, phylogenetic analysis, and transmission electron microscopy. It is worth highlighting that this study revealed the high prevalence of coinfection with YHV-8 and OWV1 in wild *P. chinensis* populations and the transmission risk of these viruses between the wild and farmed *P. chinensis* populations. This study enriches the epidemiological information of WOAH-listed and emerging diseases in wild *P. chinensis* in the Yellow Sea and raises concerns about biosecurity issues related to wild shrimp resources.

## 1. Introduction

The seed industry is a fundamental core supporting modern aquaculture. Viral diseases have been considered a vast hazard to the shrimp farming industry worldwide [1]. Wild shrimp stocks were often used for hatching and breeding to maintain the genetic diversity of farmed populations [2]. Meanwhile, the national stock enhancement program for natural fisheries resources requires artificial hatching of postlarvae from captured broodstock of local shrimp species [3], in which the interaction between the wild aquatic animal populations and aquaculture systems was highly involved. However, the epidemiological status of diseases in wild shrimp stocks is often overlooked. Pathogens, such as infectious hypodermal and hematopoietic necrosis virus (IHHNV), white spot syndrome virus (WSSV), and yellow head virus (YHV), which have caused substantial economic losses to the global shrimp farming industry, have been detected in wild black tiger shrimp *Penaeus monodon* and Indian white shrimp *P. indicus* [4,5,6,7]. Notably, coinfection with multiple pathogens is easily detected in wild shrimp populations. The individual-level coexistence of multiple pathogens in a susceptible species usually refers to coinfection with the pathogens, which may be a synergistic or antagonistic interaction between pathogens [8,9,10]. Various pathogens might have spread to oceans of the world through international trade, ocean transportation, and ocean current of water [5,11]. Shrimp can act as an asymptomatic carrier when the viral load is low and may remain infected for a long time [12,13]. However, to our knowledge, there has been little study on the coinfection or coexistence of multiple pathogens in wild Chinese white shrimp *P. chinensis*.

There are eight genotypes in yellow head complex viruses (YHVs) in the genus *Okavirus* of the family *Roniviridae*, of which three were accepted as virus species by the International Committee on Taxonomy of Viruses (ICTV). Yellow head virus (YHV; virus species *Yellow head virus*) is assigned to genotype 1 (YHV-1) [14]. Gill-associated virus (GAV; virus species *Gill-associated virus*) is assigned to genotype 2 [14]. Yellow head virus genotype 8 (YHV-8; virus species *Okavirus 1*) is assigned to genotype 8 [14]. Other genotypes (YHV-3 to YHV-7) have not been recognized as virus species by ICTV due to a lack of genome sequence [14]. Infection with YHV-1, listed as a notifiable disease by the World Organisation for Animal Health (WOAH) [15], has caused significant disease in *P. monodon* in Asia [16,17,18,19]. Infection with YHV-8 has been found to cause disease in farmed *P. chinensis* and *P. vannamei* in China and South Korea in recent years [20,21,22,23]. Oriental wenrivirus 1 (OWV1), identified from farmed *P. chinensis*, is a novel bunyavirus in the genus *Wenrivirus* of the family *Phenuiviridae* with a negative-stranded and 4-segmental RNA virus [24]. Although the virus was found in moribund *P. chinensis*, its pathogenicity is still unclear [24]. IHHNV, a member of the genus *Penstyldensovirus* in the family *Parvoviridae*, was first identified from *P. stylirostris* in Hawaii, where mortality has been notifiable to be greater than 90% [25]. Infection with IHHNV, listed as a notifiable crustacean disease by the WOAH [15], can lead to growth retardation and runt deformity syndrome [26,27,28]. Moreover, IHHNV can also be vertically transmitted [26,29]. Decapod iridescent virus 1 (DIV1) is the first virus in the new genus *Decapodiridovirus* of the family *Iridoviridae* [30]. DIV1 can infect a variety of crustaceans, resulting in significant mortality in cultured decapods and causing huge economic losses to the shrimp industry [31,32,33].

In facing the limited epidemiological information for WOAH-listed and emerging diseases in wild *P. chinensis* populations, this study collected wild *P. chinensis* individuals captured from the Yellow Sea for three consecutive years. Using molecular diagnostic methods, we confirmed the existence of IHHNV and DIV1 and coinfection with YHV-8 and OWV1 in wild *P. chinensis* populations. Consequently, the study provides epidemiological information for consideration of the biosecurity strategy related to wild *P. chinensis* resources as broodstock sources in hatching for the farmed shrimp larvae and the stock enhancement of natural resources.

## 2. Materials and Methods

### 2.1. Sample Information

Chinese white shrimp *P. chinensis* individuals of about 18–21 cm were sampled in the Yellow Sea, nine shrimp in November 2020, eight in November 2021 and twenty shrimp in March 2022. We pretreated the selected shrimp individuals on ice in advance, and all efforts were made to minimize the suffering of animals accordingly.

### 2.2. DNA and RNA Extraction

The cephalothorax was snap-frozen in liquid nitrogen to extract DNA and RNA and then used for routine pathogen detection. Total DNA and RNA were extracted from 20–30 mg individual cephalothorax tissue of *P. chinensis* by TIANamp Marine Animal DNA Kit and RNAprep pure Tissue Kit (TIANGEN Biotech, Beijing, China), respectively, according to the manufacturer’s instructions. The concentration and quality of DNA and RNA were measured by NanoDrop 2000 (Thermo Scientific, Waltham, MA, USA).

### 2.3. Pathogens Testing with Pooled Samples

We mixed DNA or RNA extracted from different batches of shrimp and detected seven pathogens, including WSSV, DIV1, IHHNV, *Enterocytozoon hepatopenaei* (EHP), covert mortality nodavirus (CMNV), infectious myonecrosis virus (IMNV), and Taura syndrome virus (TSV). Both PCR and qPCR methods targeting different sequence locations of each pathogen published in papers or recommended by the WOAH Aquatic Manual were used for molecular detection of the above seven pathogens [15,19,34,35,36,37,38,39,40,41,42,43,44,45,46,47].

### 2.4. YHVs and OWV1 Testing with Individual Samples

In order to explore whether there was coinfection with YHVs and OWV1 in wild *P. chinensis*, we detected GAV, YHV-1, YHV-8, and other YHVs by RT-PCR and OWV1 by RT-qPCR in every sampled *P. chinensis* [24,48]. When the YHVs test result is positive, the products need to be sent for Sanger sequencing, then Nucleotide-BLAST in the National Center for Biotechnology Information (NCBI) database or phylogenetic analysis to determine which genotype of YHVs. A positive control, a negative control, and a blank control were set for the PCR detection, and a positive control and a blank control were set for the qPCR detection. All primer information used in this study is listed in Table 1.

### 2.5. Transmission Electron Microscopy

For transmission electron microscopy (TEM), samples were prepared in two ways: purifying viruses from gills for negative staining and embedding gills and lymphoid organs for ultrathin sections.

Small pieces (~1 mm^3^) of the lymphoid organ and gills were sampled and fixed in the TEM fixative (2% paraformaldehyde, 2.5% glutaraldehyde, 160 mM NaCl, and 4 mM CaCl_2_ in 200 mM PBS, pH 7.2). Ultrathin sections were cut, mounted on collodion-coated grids, and stained with aqueous uranyl acetate/lead citrate using standard procedures as we previously published [24].

### 2.6. Virus Purification

Fifteen grams of *P. chinensis* gill filaments from individuals collected in 2022 were chopped and added with 100 mL of SM buffer (50 mM Tris-HCl, 10 mM MgSO_4_, 100 mM NaCl, pH 7.5) with 0.5 mM 4-(2-Aminoethyl) benzenesulfonyl fluoride hydrochloride (AEBSF) (Solarbio, Beijing, China). The tissue was homogenized by a homogenizer in an ice bath at a speed of 10,000 rpm for 5 s. These steps were repeated several times (> 3) until a uniform tissue homogenate was obtained. The homogenate was centrifuged at 1400 g for 15 min at 4 °C (CR21GIII, Hitachi, Tokyo, Japan). A 20 mL of SM buffer was added to the pellet and homogenized in an ice bath at 10,000 rpm for 10 s, followed by centrifugation at 6000 g for 15 min at 4 °C. These two supernatants obtained in the above steps were combined, filtered through a 38 μm mesh, and the filtrate was centrifuged at 10,000 g for 25 min at 4 °C. The final supernatant was again filtered through a 38 μm mesh, mixed with an equal volume of SM buffer, and centrifuged at 40,000 g for 2 h at 4 °C (CP100WX; Hitachi, Tokyo, Japan) to pellet viral particles [49]. Finally, 500 μL of SM buffer was added to the pellet to suspend it.

The supernatant and the pellet suspension were dropped on grids, negatively stained with 2% phosphotungstic acid (pH 6.5, Solarbio, Beijing, China), and observed under a TEM (HT7700, Hitachi, Tokyo, Japan) at 80 kV.

### 2.7. Phylogenetic Analyses

For the genus *Okavirus*, the approach proposed by Mohr was employed using primers YH30 m/31 m to get a partial sequence of the open reading frame 1b (ORF1b) region to clarify the genotype [19]. For the genus *Wenrivirus,* we constructed four pairs of primers (Table 2) with overlapping regions for the virus and obtained the full length of the RNA-directed RNA polymerase (RdRp) of the virus via Geneious version 2022.1.1 (Biomatters Ltd., Auckland, New Zealand) [50] splicing to obtain the sequence of the RdRp region of OWV1 in the sample. Then we used Geneious version 2022.1.1 to translate the obtained nucleotide sequences into amino acid sequences. For determining the evolutionary relationship of *Wenrivirus* in the samples collected in this paper, the entire RdRp region of *Wenrivirus* and some *Phenuiviridae* viruses were selected (Refer to another published article for virus names, etc. [24]). Multiple sequence alignments of *Okavirus*-related nucleic acid sequences were performed using the MUSCLE program in the MEGA version 11.0.10 software [51], and “Find Best DNA Models” were used to determine the most suitable models for *Okavirus*. Then based on the lowest Bayesian information criterion score, the Tamura-Nei model with discrete gamma distribution (TN93+G) was determined as the best model for *Okavirus*. Then based on the lowest Bayesian information criterion score, the LG with frequencies model gamma distributed with invariant sites (LG+G+I+F) was determined as the best model for *Wenrivirus.* The maximum likelihood phylogenetic tree was constructed using the best model for *Okavirus* and *Wenrivirus*. Phylogenetic testing was performed using the bootstrap method with 1000 replicates. All relevant sequences have been submitted to NCBI.

## 3. Results

### 3.1. Testing Results of Pooled Samples

To understand the pathogen information in wild *P. chinensis* populations, we detected the pooled samples for seven known pathogens, including DIV1, IHHNV, CMNV, EHP, WSSV, IMNV, and TSV. The PCR and qPCR results (Table 2) show that all shrimp samples collected in three years were negative for five pathogens, including CMNV, EHP, WSSV, IMNV, and TSV. The samples collected in 2020 showed DIV1 and IHHNV positive, and the samples collected in 2022 had IHHNV positive as well.

### 3.2. Testing Results of Individual Samples for YHVs and OWV1

All individual samples collected in three years were positive for two pathogens, including OWV1 and YHVs (Appendix A, Table 3). The Ct value of the individual OWV1 detection in three years changed from 9.2 ± 0.1 to 35.0 ± 1.7 cycles, in which the batch average Ct value of OWV1 was 19.2 ± 7.6, 16.3 ± 6.6, and 25.0 ± 6.7 in 2020, 2021, and 2022 (with significant differences to 2020 and 2021), respectively. After sequencing and BLAST in the NCBI database, we confirm that all YHVs positives are YHV-8.

### 3.3. TEM Examination

TEM performed with ultrathin sections of lymphoid organs and gills of *P. chinensis* showed spherical to oval virus particles (80 nm–115 nm) of OWV1 in the cytoplasm of the gill tissue (Figure 1A,B). However, no YHVs particle was observed in the gill and lymphoid organ tissues. Alternatively, the supernatant and pellet suspension from virus purification were observed with TEM. The results showed a few rod-like structures in the supernatant, with multiple linear structures but no envelope similar to our previous observation [49] (Figure 1C,D).

### 3.4. Homology and Phylogenetic Analyses

To unambiguously identify the genotype of YHVs, we performed a phylogenetic analysis of the samples collected in this study. The products of the YHV ORF1b region obtained were sent for sequencing. After NCBI Nucleotide-BLAST (Table 4), it shows that the ORF1b part of the YHV isolates collected in this study had a 98.32–99.24% similarity to the corresponding segment of YHV-8 20120706 (NC_048215.1). Furthermore, the results of the phylogenetic analysis show that YHVs sequenced in this study cluster with YHV-8 previously isolated from farmed *P. chinensis* populations [22] and are separated from the other seven genotypes (Figure 2). Therefore, YHV-8 was identified from the wild *P. chinensis* collected in this study.

The obtained product of OWV1 RdRp region was sent for sequencing and translated into the amino acid sequence using Geneious version 2022.1.1. After NCBI Protein-BLAST (Table 5), it can be seen that the RdRp of the OWV1 isolates collected in this study had a 99.14%–99.57% similarity to the corresponding segment of OWV1 (QHW05228.1). Moreover, the phylogenetic analysis results show that OWV1 isolates of wild *P. chinensis* cluster with the OWV1 reference sequence previously isolated from farmed *P. chinensis* [24] and then gather with Wenzhou shrimp virus 1 (WzSV-1) and Mourilyan virus (MoV) into the genus *Wenrivirus* (Figure 3).

## 4. Discussion

Shrimp diseases have become an important factor hindering the shrimp industry’s green development of the shrimp farming industry [52]. This study revealed that multiple pathogens, including YHV-8, OWV1, DIV1, and IHHNV, can be detected in wild *P. chinensis* populations in the Yellow Sea. Furthermore, we observed coinfection with YHV-8 and OWV1 in all *P. chinensis* individuals by molecular diagnosis, phylogenetic analysis, and TEM. As the *P. chinensis* farming industry still largely relays on the wild *P. chinensis* broodstock [3], the consistently high prevalence of YHV-8 and the existence of DIV1 in wild *P. chinensis* reveal significant risks in the shrimp farming industry and the stock enhancement program for fishery resources of *P. chinensis* in East Asia.

Unlike human health, surveillance for aquatic animal health usually emphasizes the population-based approach. Detection of a lower prevalence in a larger population requires a larger sample size [53], which requires more tests. Therefore, pooling individual samples before a diagnostic test is a commonly used strategy. WOAH standard recommends a pooling rate of less than 5:1 [15], which may still result in a large number of tests for a large population. Our previous study using the TaqMan-qPCR method to evaluate the pooling rate revealed that a 50:1 pooling rate could have a similar diagnostic sensitivity to the pooling rate of 5:1 [54]. All the pooling modes from 5:1 to 150:1 have good diagnostic specificity. The tests of our present study using a pooling rate of no more than 20:1 should not significantly impact the diagnostic specificity and sensitivity.

Compared with the farmed samples, the accessibility of captured wild *P. chinensis* is much more difficult due to the declined fishery resources and uncertainty of seasonal migration caused by climate change [55,56]. Meanwhile, weak and diseased shrimp individuals may much easier be predated by carnivorous fishes or sink to the deep bottom. In facing these challenges, we used two different PCR methods targeting two sequence locations for each of the seven pathogens except YHVs and OWV1. This strategy can verify false results due to low specificity, low sensitivity, and contamination in sample preparation or PCR methods. Nevertheless, the PCR and qPCR results for each pathogen were highly coincident, which indicated that the molecular results were reliable.

Unlike the research on the viral diseases of farmed crustaceans, the reports on the spread of wild crustacean viruses and coinfection with multi-pathogen are still very limited [7]. Spann et al. [57] reported that coinfection with MoV and GAV is very common in diseased *P. monodon*. Notably, YHV-8 and GAV belong to *Okavirus*, and MoV and OWV1 belong to *Wenrivirus*. The similarity of the virus taxonomy and the coinfection in Spann et al.’s and our results raise the interesting issue of whether the *Okavirus* and *Wenrivirus* have a synergistic mechanism in the concomitance of coinfection. Teixeira-Lopes et al. [58] revealed that viral coinfection could regulate the viral load in the host, and there may be a negative correlation between the two viruses when they coexist. On the other hand, a number of studies reported that shrimp previously infected with IHHNV gained resistance to infection with WSSV and significantly reduced the mortality of shrimp. However, the survival rate of shrimp infected with TSV and YHV was significantly higher than that of shrimp infected only with YHV [10,59]. Further study on interactions between concomitant YHV-8 and OWV1 in *P. chinensis* populations may help to discover the deep infection mechanism of shrimp viruses.

The confirmatory diagnosis of a suspected case of infection with a specific pathogen is generally based on meeting the criterium of two or more independent tests in the WOAH Manual [15]. The diagnosis of infection with DIV1 also follows the same principle [60]. With qPCR and nested PCR, this study confirmed that wild *P. chinensis* in the Yellow Sea has been infected with IHHNV and DIV1. IHHNV natively originated from the Western Hemisphere. It was first detected from farmed *P. vannamei* in 2001 in China [61]. *P. chinensis* has not been listed as a susceptible species of IHHNV [15]. However, IHHNV positives have been detected from farmed *P. chinensis* in the National Target Surveillance Program of China in 2019 [62]. IHHNV detected in wild populations of *P. chinensis* implies the transmission risk of viruses between farmed and wild shrimp populations. DIV1 was detected in the wild *P. monodon* captured from the northeastern Indian Ocean [6]. DIV1 detected from wild *P. chinensis* indicated a broader virus spread in the wild. We did not find viral particles like DIV1 or IHHNV with TEM, which may be related to the selected tissues, centrifugal force, and low viral load. After subsequent virus purification, a large number of suspected nucleocapsid complex structures, likely YHV-8 virions, were observed in the supernatant. However, no complete complex structure was found, which might be caused by a large centrifugal force [49].

*Okavirus* isolates from captured wild *P. chinensis* cluster with YHV-8 previously identified from farmed *P. chinensis* [22] and separated from the other seven genotypes in the phylogenetic analysis. Similarly, the phylogenetic analysis of OWV1 showed that OWV1 isolates from wild and farmed shrimp [24] cluster together. These phylogenetic analyses provide information for tracing the source of YHV-8 and OWV1 infection in wild and farmed *P. chinensis*. Conclusively, the sequence similarity of the isolates from farmed and wild *P. chinensis* for both viruses implies the transmission risk of viruses between farmed and wild shrimp populations.

This study has detected multiple pathogens in wild *P. chinensis*, which reminds pathogenic risks in wild shrimp populations are nonnegligible. Wild shrimp used for broodstock must be strictly inspected and quarantined to select specific pathogen-free stocks [4,29,63]. Diseased and virus-carrying wild shrimp should be strictly prevented from entering the aquaculture systems. The national surveillance plan targeting specific pathogens may need to be extended to wild shrimp [64]. On the other hand, hatcheries for stock enhancement should implement strictly high-quality biosecurity measures to ensure pathogens from aquaculture systems will not be released into wild populations [3]. Implementation of biosecurity measures is critical not only for preventing disease risks in shrimp farms but also for securing the ecosystem of wild shrimp populations.

## 5. Conclusions

This study is the first to report the coexistence of multiple viruses, including DIV1, IHHNV, YHV-8, and OWV1. Notably, the results reveal a consistently high prevalence of coinfection with YHV-8 and OWV1 in wild *P. chinensis* populations and the transmission risk of viruses between wild and farmed populations. The findings imply that we should pay attention to the epidemiology of diseases in wild *P. chinensis*, implement disease surveillance on wild shrimp, and introduce biosecurity policies and measures to prevent disease risks both in shrimp farms and hatcheries for stock enhancement.

## Figures and Tables

**Figure 1 viruses-15-00361-f001:**
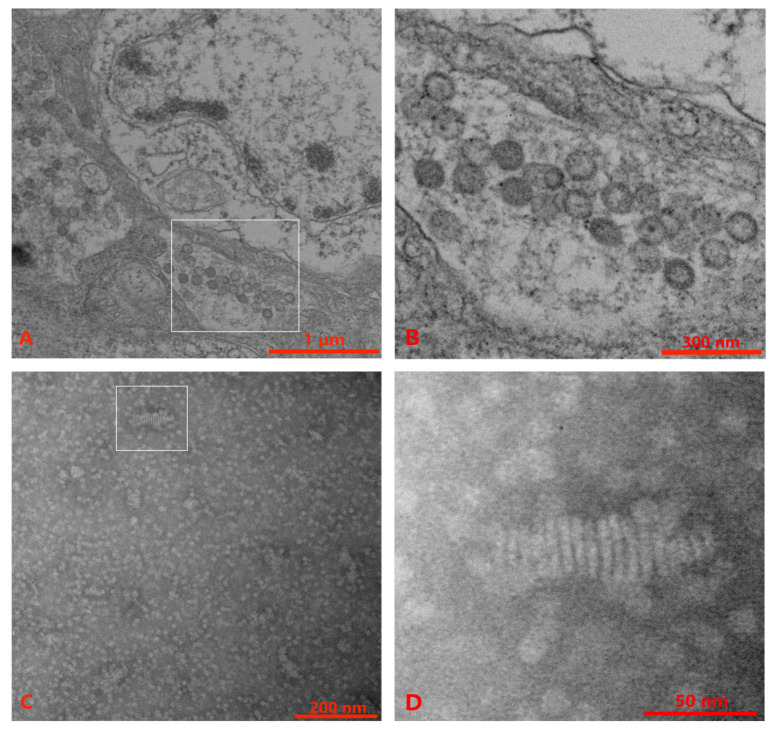
Transmission electron microscopy (TEM) observations of wild *P. chinensis*. (**A**) Gill tissue with sample number 20211126007. (**B**) High magnification of the gill tissue section reveals abundant oriental wenrivirus 1 (OWV1) virions. (**C**) The supernatant was obtained after virus purification. (**D**) High magnification shows the obvious rod-like and multiple-linear structure with no envelope. Scale bars = (**A**) 1 μm, (**B**) 300 nm, (**C**) 200 nm, (**D**) 50 nm.

**Figure 2 viruses-15-00361-f002:**
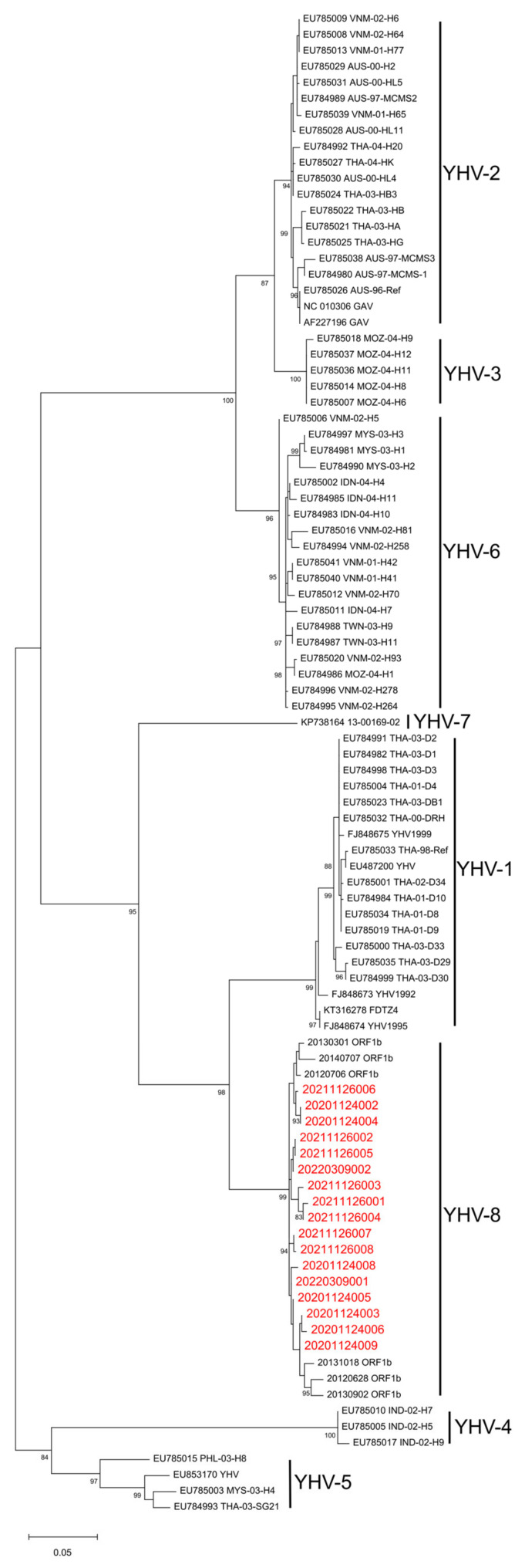
Phylogenetic tree analysis of partial open reading frame 1b (ORF1b) region of yellow head complex viruses (YHVs). The maximum likelihood evolutionary trees of YHVs were constructed using MEGA version 11.0.10 software. The red font represents the samples collected in this experiment. All reference sequences are downloaded from GenBank. Bootstrap values were calculated with 1000 replicates of the alignment. Percentage bootstrap values (1000 replicates) >80% are shown. The partial nucleotide sequences of ORF1b from 20201124002, 20201124003, 20201124004, 20201124005, 20201124006, 20201124008, 20201124009, 20211126001, 20211126002, 20211126003, 20211126004, 20211126005, 20211126006, 20211126007, 20211126008, 20220309001 and 20220309002 have been submitted to the GenBank databases under accession number OP902176, OP902177, OP902178, OP902179, OP902180, OP902181, OP902182, OP902185, OP902186, OP902187, OP902200, OP902201, OP902202, OP902203, OP902204, OP902183 and OP902184, respectively.

**Figure 3 viruses-15-00361-f003:**
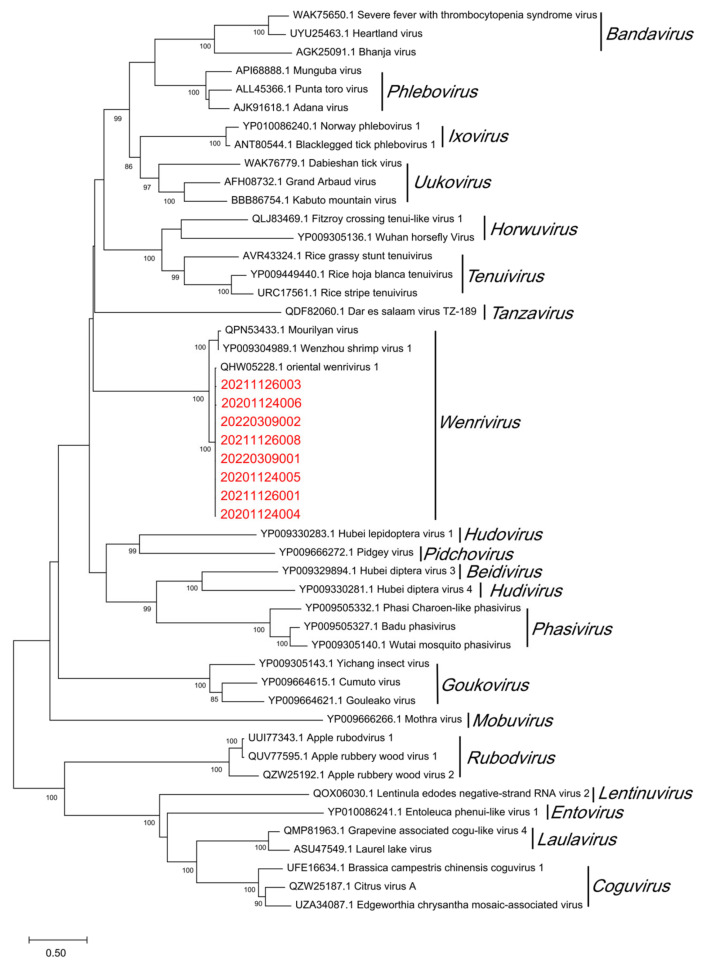
Phylogenetic tree of the RNA-directed RNA polymerase (RdRp) protein sequences of oriental wenrivirus 1 (OWV1) and some viruses in the family *Phenuiviridae*. The maximum likelihood evolutionary trees of *Phenuiviridae* were constructed using MEGA version 11.0.10 software. The red font represents the samples collected in this experiment. All reference sequences are downloaded from GenBank. Bootstrap values were calculated with 1000 replicates of the alignment. Percentage bootstrap values (1000 replicates) >80% are shown. The partial nucleotide sequences of RdRp from 20201124004, 20201124005, 20201124006, 20211126001, 20211126003, 20211126008, 20220309001 and 20220309002 have been submitted to the GenBank databases under accession number OP902199, OP902198, OP902197, OP902196, OP902194, OP902204, OP902189 and OP902188, respectively.

**Table 1 viruses-15-00361-t001:** Primer sequences were used in this study.

Primer Name	Sequence (5′-3′)	Source
WSS-1011F	TGGTCCCGTCCTCATCTCAG	[35]
WSS-1079R	GCTGCCTTGCCGGAAATTA	
WSS-Probe	AGCCATGAAGAATGCCGTCTATCACACA	
WSSV-146F1	ACTACTAACTTCAGCCTATCTAG	[34]
WSSV-146R1	TAATGCGGGTGTAATGTTCTTACGA	
WSSV-146F2	GTAACTGCCCCTTCCATCTCCA	
WSSV-146R2	TACGGCAGCTGCTGCACCTTGT	
DIV1-142F	AATCCATGCAAGGTTCCTCAGG	[47]
DIV1-142R	CAATCAACATGTCGCGG GAAC	
DIV1-Probe	CCATAGTGCTCGCTCGGCTTCGG	
DIV1-F1	GGGCGGGAGATGGTGTTAGAT	[31]
DIV1-R1	TCGTTTCGGTACGAAGATGTA	
DIV1-F2	CGGGAAACGATTCGTATTGGG	
DIV1-R2	TTGCTTGATCGGCATCCTTGA	
IHHNV-1608F	TACTCCGGACACCCAACCA	[36]
IHHNV-1688R	GGCTCTGGCAGCAAAGGTAA	
IHHNV-Probe	ACCAGACATAGAGCTACAATCCTCGCCTATTTG	
IHHNV-389F	CGGAACACAACCCGACTTTA	[38]
IHHNV-389R	GGCCAAGACCAAAATACGAA	
EHP-157F	AGTAAACTATGCCGACAA	[37]
EHP-157R	AATTAAGCAGCACAATCC	
EHP-Probe	TCCTGGTAGTGTCCTTCCGT	
EHP-SWP-1F	TTGCAGAGTGTTGTTAAGGGTTT	[39]
EHP- SWP-1R	CACGATGTGTCTTTGCAATTTTC	
EHP-SWP-2F	TTGGCGGCACAATTCTCAAACA	
EHP-SWP-2R	GCTGTTTGTCTCCAACTGTATTTGA	
CMNV-TAQ-F	CGAGCTAATCCAAGCACTTC	[42]
CMNV-TAQ-R	ACCTGTTAGGTACGCTACCA	
CMNV-TAQ-Probe	CGCTCACGGCTTTGGAT ACCTT	
CMNV-7F1	AAATACGGCGATGACG	[44]
CMNV-7R1	ACGAAGTGCCCACAGAC	
CMNV-7F2	CACAACCGAGTCAAACC	
CMNV-7R2	GCGTAAACAGCGAAGG	
IMNV-412F	GGACCTATCATACATAGCGTTGCA	[40]
IMNV-545R	AACCCATATCTATTGTCGCTGGAT	
IMNV-Probe	CCACCTTTACTTTCAATACTACATCATCCCCGG	
IMNV-4587F	CGACGCTGCTAACCATACAA	[43]
IMNV-4914R	ACTCGGCTGTTCGATCAAGT	
IMNV-4725NF	GGCACATGCTCAGAGACA	
IMNV-4863NR	AGCGCTGAGTCCAGTCTTG	
TSV-1004F	TTGGGCACCAAACGACATT	[46]
TSV-1075R	GGGAGCTTAAACTGGACACACTGT	
TSV-Probe	CAGCACTGACGCACAATATTCGAGCATC	
TSV-7171F	CGACAGTTGGACATCTAGTG	[45]
TSV-7511R	GAGCTTCAGACTGCAACTTC	
OWV1-S1-F	CCGACATGGATGCGTTCA	[24]
OWV1-S1-R	CAAGGCTGCAACAATGACCTT	
OWV1-S1-Probe	CGCAGACATCCAGTTCCAGGGCTTT	
OWV1-F1	CAAAGACGGGTGTTGTAGTGT	In this study
OWV1-R1	AGCTTGCCTCACCGTTCTC	
OWV1-F2	ATTCGTCTCACTCCTGGCTG	
OWV1-R2	CGCCTTCCATGTACTCGCTA	
OWV1-F3	CCACCACCTCAAGTCCACA	
OWV1-R3	TCATCAGAGTATGGGGACAGGT	
OWV1-F4	TGAAGAGGAGGGAGGTGCAT	
OWV1-R4	ACAAAGACCGGGTGTGTTCTAA	
GY1	GACATCACTCCAGACAACATCTG	[48]
GY2	CATCTGTCCAGAAGGCGTCTATGA	
GY4	GTGAAGTCCATGTGTGTGAGACG	
GY5	GAGCTGGAATTCAGTGAGAGAACA	
Y3	ACGCTCTGTGACAAGCATGAAGTT	
G6	GTAGTAGAGACGAGTGACACCTAT	
YH30-F1m	TACCAYTCAAACATCATYAAYAAYCAYCA	[19]
YH30-R1m	GAGATGATYTGRTKCTTRAAYTTCTGRAA	
YH31-F2m	CTCARATCCATGCMATYTGGGARTCHTC	
YH31-R2m	AGTTTGGCRCGRATRTTRGTRAGRAT	

**Table 2 viruses-15-00361-t002:** Testing results for seven pathogens with the pooled samples.

Viruses	Species	Detection Method	Number of Samples		
			20201124001-009	20211126001-008	20220309001-020
DIV1	*P. chinensis*	Nested -PCR	P	N	N
		qPCR (cycles)	31.3 ± 4.3	N	N
IHHNV	*P. chinensis*	PCR	P	N	P
		qPCR (cycles)	36.7 ± 1.4	N	28.5 ± 0.2
CMNV	*P. chinensis*	Nested RT-PCR	N	N	N
		RT-qPCR (cycles)	N	N	N
EHP	*P. chinensis*	Nested -PCR	N	N	N
		qPCR (cycles)	N	N	N
WSSV	*P. chinensis*	Nested -PCR	N	N	N
		qPCR (cycles)	N	N	N
IMNV	*P. chinensis*	Nested RT-PCR	N	N	N
		RT-qPCR (cycles)	N	N	N
TSV	*P. chinensis*	RT-PCR	N	N	N
		RT-qPCR (cycles)	N	N	N

“*P. chinensis*” stands for *Penaeus chinensis*. “cycles” stands for threshold cycles. “20201124001-009” stands for 9 wild *P. chinensis* sampled in 2020. “20211126001-008” stands for 8 wild *P. chinensis* sampled in 2021. “20220309001-020” stands for 20 wild *P. chinensis* sampled in 2022. “P” stands for positive. “N” stands for negative.

**Table 3 viruses-15-00361-t003:** Oriental wenrivirus 1 (OWV1) and yellow head complex viruses (YHVs) testing results of each sample.

Number of Samples	Species	Viruses and Detection Method	
		OWV1 RT-qPCR (Cycles)	YHVs Nested RT-PCR
20201124001	*P. chinensis*	13.8 ± 0.1	P
20201124002	*P. chinensis*	25.2 ± 1.0	P
20201124003	*P. chinensis*	11.4 ± 0.1	P
20201124004	*P. chinensis*	12.3 ± 0.2	P
20201124005	*P. chinensis*	11.9 ± 0.3	P
20201124006	*P. chinensis*	13.2 ± 0.2	P
20201124007	*P. chinensis*	29.1 ± 0.1	P
20201124008	*P. chinensis*	26.6 ± 0.1	P
20201124009	*P. chinensis*	29.6 ± 0.1	P
20211126001	*P. chinensis*	26.2 ± 0.1	P
20211126002	*P. chinensis*	10.2 ± 0.2	P
20211126003	*P. chinensis*	19.6 ± 0.1	P
20211126004	*P. chinensis*	26.6 ± 0.1	P
20211126005	*P. chinensis*	11.3 ± 0.6	P
20211126006	*P. chinensis*	12.6 ± 0.4	P
20211126007	*P. chinensis*	9.2 ± 0.1	P
20211126008	*P. chinensis*	14.5 ± 0.2	P
20220309001	*P. chinensis*	20.2 ± 0.3	P
20220309002	*P. chinensis*	25.0 ± 0.8	P
20220309003	*P. chinensis*	26.2 ± 0.4	P
20220309004	*P. chinensis*	14.1 ± 0.5	P
20220309005	*P. chinensis*	19.1 ± 0.9	P
20220309006	*P. chinensis*	19.0 ± 1.1	P
20220309007	*P. chinensis*	32.5 ± 3.2	P
20220309008	*P. chinensis*	13.7 ± 1.5	P
20220309009	*P. chinensis*	28.6 ± 0.5	P
20220309010	*P. chinensis*	30.4 ± 0.0	P
20220309011	*P. chinensis*	27.3 ± 0.7	P
20220309012	*P. chinensis*	29.9 ± 1.1	P
20220309013	*P. chinensis*	24.2 ± 0.4	P
20220309014	*P. chinensis*	26.9 ± 0.5	P
20220309015	*P. chinensis*	31.2 ± 0.8	P
20220309016	*P. chinensis*	35.0 ± 1.7	P
20220309017	*P. chinensis*	10.0 ± 6.5	P
20220309018	*P. chinensis*	32.4 ± 3.5	P
20220309019	*P. chinensis*	29.1 ± 0.9	P
20220309020	*P. chinensis*	25.4 ± 0.1	P

**Table 4 viruses-15-00361-t004:** Alignment of the open reading frame 1b (ORF1b) nucleotide sequence of yellow head virus genotype 8 (YHV-8) isolated from wild *P. chinensis* with the corresponding sequence of YHV-8 20120706 previously isolated from farmed *P. chinensis*.

Source	Object of Comparison	Max Score	Query Cover	Percent Identity
20201124002	YHV-8 20120706 (NC_048215.1)	1203	97%	99.10%
20201124003	YHV-8 20120706 (NC_048215.1)	1186	97%	98.65%
20201124004	YHV-8 20120706 (NC_048215.1)	1197	96%	99.10%
20211124005	YHV-8 20120706 (NC_048215.1)	1203	99%	98.96%
20201124006	YHV-8 20120706 (NC_048215.1)	1186	97%	98.65%
20201124008	YHV-8 20120706 (NC_048215.1)	1197	97%	98.95%
20201124009	YHV-8 20120706 (NC_048215.1)	1192	96%	98.80%
20211126001	YHV-8 20120706 (NC_048215.1)	1151	100%	98.32%
20211126002	YHV-8 20120706 (NC_048215.1)	1168	100%	98.78%
20211126003	YHV-8 20120706 (NC_048215.1)	1151	100%	98.32%
20211126004	YHV-8 20120706 (NC_048215.1)	1162	100%	98.63%
20211126005	YHV-8 20120706 (NC_048215.1)	1168	100%	98.78%
20211126006	YHV-8 20120706 (NC_048215.1)	1184	100%	99.24%
20211126007	YHV-8 20120706 (NC_048215.1)	1175	99%	99.08%
20211126008	YHV-8 20120706 (NC_048215.1)	1170	99%	98.93%
20220309001	YHV-8 20120706 (NC_048215.1)	1203	97%	99.10%
20220309002	YHV-8 20120706 (NC_048215.1)	1197	97%	98.95%

**Table 5 viruses-15-00361-t005:** Alignment of the RNA-directed RNA polymerase (RdRp) amino acid sequence of oriental wenrivirus 1 (OWV1) isolated from wild *P. chinensis* with the corresponding sequence of OWV1 previously isolated from farmed *P. chinensis*.

Source	Object of Comparison	Max Score	Query Cover	Percent Identity
20201124004	OWV1 (QHW05228.1)	1449	100%	99.42%
20201124005	OWV1 (QHW05228.1)	1451	99%	99.57%
20201124006	OWV1 (QHW05228.1)	1446	100%	99.14%
20211126001	OWV1 (QHW05228.1)	1449	100%	99.42%
20211126003	OWV1 (QHW05228.1)	1447	100%	99.14%
20211126008	OWV1 (QHW05228.1)	1449	100%	99.42%
20220309001	OWV1 (QHW05228.1)	1451	100%	99.57%
20220309002	OWV1 (QHW05228.1)	1449	100%	99.42%

## Data Availability

All data are available upon request.

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
