# Peer review of "Coinfection with Yellow Head Virus Genotype 8 (YHV-8) and Oriental Wenrivirus 1 (OWV1) in Wild Penaeus chinensis from the Yellow Sea"

_viruses, 2023, doi:10.3390/v15020361_

Round 1

Reviewer 2 Report

This manuscript addresses the concern of transmission of virus between wild and farmed shrimp (Penaeus chinensis) and the discuss the potential impact of co-infections of viruses in Penaeus chinensis with a special focus on YHOV-8 and OWV1 in wild P. chinensis. The information is new and deserves to be published, but the manuscript could be considered as to be restructured to a short communication. Further, a few minor comments/questions must be addressd before publication. Also, the paper would greatly benefit from improvement of the language.

Introduction:

Is description of the virus structure of importance? Perhaps is a description of the disease the viruses may cause and the dissemination of them, of greater interest.

Use the terminology “infection” and “disease” correctly consistently throughout the paper.

IHHNV: first stated that mortality can be greater than 90%, then in line 62 it is stated that it does not cause mortalities.. please clarify.

Line 64: “parent shrimp” is redundant in this context.

M and M:

Please specify how the animals were killed.

Table 1: Change “number of samples” (internal sample ID?), “species” and “location” could be part of the ledgend. The table could be removed and easily be integrated in the text. For example “ P.chinesis of about 18-21 cm were sampled in Yellow Sea; Nine fish in November 2020, eight in November 2021 and 20 fish in March 2022”.

2.4: virus purification and 2.5 TEM: Explain the purpose of the purification and the different preparation methods of the tissue, e.i. an introduction to this paragraph could be: “Samples for TEM were prepared in two ways; purification of virus from gills and by imbedding gills and lymphoid organs enabling to visualize the virus in situ

What controls are included in the different experiments? Please clarify in the text

Results:

3.1: This paragraph is very confusing. Firstly; it is stated that all 9 pathogens were detected (line 142), but, then in line 149, “…(5 agents) were not detected”?. Secondly, the paragraph must be structured, the information is also given in table 3, and by referring to the table in the text, the results will be clearer.  What do the authors mean by “about 27.04…?” this is not “about”, but very exact. Thirdly, figure 1 is redundant (could be part of supplementary?), and legend for table 3 is lacking all relevant information (abbreviations, number of samples, use 9, 8 and 20 instead of the numbers, perhaps just include the year of sampling, P (positive), N (negative).

3.2: TEM: I think the image would benefit from some white balance adjustment and the scale bars should be more easy to see.
